# Inhibition of complement C3 prevents osteoarthritis progression in guinea pigs by blocking STAT1 activation
Jen X. Xu [1] ✉, Frank Z. Xu[1,3], Amelia Furbish[1], Alicia M. Braxton [2], Brook Brumfield[1], Kristi L. Helke [2] & Yuri K. Peterson [1] ✉

Osteoarthritis (OA) is one of the leading causes of disability, affecting over 500 million adults worldwide. Previous studies have found that various inflammatory factors can contribute to the pathogenesis of OA, including complement factors in the synovial fluid of OA patients. However, the pathogenesis of this disease is still not known, and the only therapy of severe OA is total joint replacements. Total joint replacements are invasive, expensive, and affect quality of life. Here we show that when human articular chondrocytes are stimulated with pro-inflammatory mediator interleukin-1β (IL-1β) there is an increase in inflammatory factors including complement component 3 (C3). We also found the transcription factor, signal transducer and activator of transcription 1 (STAT1), is responsible for increased C3 expression after IL-1β stimulation in human articular chondrocytes. A specific STAT1 inhibitor, fludarabine, attenuates the hyper-expression of C3 and delays/prevents spontaneous OA in Dunkin-Hartley guinea pigs. Since fludarabine is already clinically used for chemotherapy, this study has great translational potential as a unique disease-modifying osteoarthritis drug (DMOAD) in treating primary OA.

Primary osteoarthritis (OA) is the most common cause of knee arthritis worldwide[1]. OA leads to chronic disability due to pain and joint dysfunction[2]. Joint replacement for end-stage OA remains the only remedy, leading to an escalating healthcare problem in our obese and ageing society[3]. OA is characterized by articular cartilage damage and loss, together with structural abnormalities of subchondral bone and low-grade chronic joint inflammation[2,4]. There are no effective treatments to prevent the progression of OA. It is unknown which processes trigger this disease or the inflammation that is associated with it[4]. Current studies discovered that OA involves crosstalk between the synovium, articular cartilage, and subchondral bone, although the causes of cartilage degradation remain uncertain[2,3]. Despite the profound clinical and socioeconomic impacts of OA, our understanding of its mechanisms is unclear[1,4]. Accelerating drug target discoveries will increase understanding of joint physiology and disease pathogenesis, which will facilitate new treatments that will prevent joint and cartilage destruction in OA pathogenesis.

Mild wear and tear may initiate certain inflammatory and cell regulatory cascades, which cause cartilage loss that eventually leads to joint damage[5]. Several studies have found increased acute inflammatory factors, including complement factors, in the synovial fluid of arthritic joints and linked these factors to the pathogenesis of the disease[6,7]. The inflammatory process in OA starts with focal inflammatory factors and cytokines but fewer typical inflammatory cells, unlike other inflammatory diseases, such as autoimmune arthritis (rheumatoid, lupus, psoriasis, etc.) and infections where there are higher level of neutrophils, macrophages, and other inflammatory cells[8]. Our studies focus on finding a key molecule/pathway in OA inflammation pathways in order to develop a novel DMOAD for primary (spontaneous) OA. Complement proteins are part of the innate immune system and are mainly made by the liver[6,9]. They normally circulate the plasma and can be found in the synovium as well[6,7]. Complement proteins are activated via a variety of triggers through three major pathways- classical, lectin, and alternative. All three of these pathways converge on the key step of complement component 3 (C3) cleavage by C3 convertase[6,10]. C3 convertase has a variety of effector functions, including inflammation, formation of the membrane attack complex (MAC), and opsonization[11]. C3 convertase converts C3 into its active components, C3a and C3b[9]. C3a is involved in inflammation, and C3b is involved in both opsonization and converting complement factor 5 (C5) into its active

[1]Department of Drug Discovery and Biomedical Sciences, Medical University of South Carolina, 70 President Street, Charleston, SC 29425, USA. [2]Department of Comparative Medicine, Medical University of South Carolina, 114 Doughty Street, Charleston, SC 29425, USA. [3]Present address: UAB Heersink School of Medicine, Alabama, AL 35233, USA. ✉e-mail: xuje@musc.edu; petersy@musc.edu

components[6]. Activated factor B (Bb) will combine with C3b to form C3 convertase (C3bBb) through the alternative pathway[12]. Another important function of C3 convertase is the amplification of the complement cascade which accounts for the intense, innate inflammatory response the complement system can cause[10]. Previous studies have identified local production of complement factors by synovial cells is responsible for joint degradation of OA instead of systemic production and circulation of complement factors as the joint space is an isolated environment and particles mainly pass via diffusion[13,6]. Oral medications, such as analgesics or non-steroidal anti-inflammatory drugs (NSAIDs), can cause side effects and have contraindications in elderly patients, which are the largest demographic for OA. Additionally, oral medications are not specific and cannot easily diffuse into the joint space. Currently, there is no DMOAD that is approved by the United States Food and Drug Administration or the European Medicines Agency[14,15]. Most investigations are currently focused on intra-articular (IA) injection of cellular therapies such as human serum albumin, metformin, statins, cytokines and growth factors antagonists, bone morphogenetic protein, platelet-rich plasma and mesenchymal stem cells[16]. However, none of these delay the progression of primary OA. Glucocorticoids such as cortisone or triamcinolone acetonide are the most commonly used agents for IA injection to provide OA patients with short-term pain reduction and improvement in function[17]. Steroids, like glucocorticoids, are anti-inflammatory, and block many different inflammatory pathways, such as IL-6, IL-8, NF-κB, STAT3, collagen I, MMP-1, and MMP-13, which lead to a plethora of side effects[18,19]. However, IA injections of glucocorticoids not only show a lack of improvement after three months to two years, but also result in significant cartilage loss, increase risk of septic arthritis, postoperative joint infection, and long-term negative impacts on joint health[20,21].

Our study identified C3 to be a key element in the inflammatory process during primary OA development. In addition, signal transducer and activator of transcription 1 (STAT1) is responsible for the initial early increase in C3 expression after primary chondrocytes were stimulated with IL-1β. Therefore, we chose STAT1 as a therapeutic target for treating the development of spontaneous OA in guinea pigs. Dunkin-Hartley guinea pigs spontaneously develop OA starting at three-months old[22]. We found that fludarabine, a specific STAT1 inhibitor, inhibited the expression of C3, and protected the knee joints from cartilage/bone damage in spontaneous OA in guinea pigs.

## Results

### Interleukin-1β (IL-1β) stimulation increased expression of complement factors in human primary articular chondrocytes

To mimic the pathogenesis of osteoarthritis in the joint space of humans for our in vitro OA model, we stimulated human primary articular chondrocytes with IL-1β. IL-1β is present in OA patients' synovial fluid, mimics the process of inflammation in OA, and is commonly used as an in vitro model as previously reported[23]. The ELISA results showed that IL-6 is highly expressed in IL-1β treated chondrocytes, with or without growth serum (Fig. S1a). This result is consistent with other studies that showed increased IL-6 expression in the synovial fluid, articular cartilage, and infrapatellar fat pad of patients with OA[24,25].

To further investigate what factors are responsible for the increased inflammation, we performed liquid chromatography coupled with mass spectrometry (LC-MS/MS) of the supernatant of human primary articular chondrocytes grown without growth serum. LC-MS/MS revealed several protein factors had increased peptide coverage in the cells grown with IL-1β compared to control chondrocytes (Supplementary Data 1). A group of peptides that were elevated in chondrocytes treated with IL-1β compared to chondrocytes without IL-1β were extracellular matrix proteins including fibronectin, collagen, vascular cell adhesion molecule 1 (VCAM1), and matrix metalloproteinases (MMPs). Previous studies showed MMP inhibitors for OA treatment failed in clinical trials due to musculoskeletal toxicity[26,27]. We also identified several complement factors that were markedly increased in the cells treated with IL-1β, specifically C3 and factor

B. In the alternative pathway, C3 and factor B levels will come together and form C3 convertase (C3bBb). Therefore, it prompted us to focus on inflammatory factors for drug discovery in OA. We also observed that complement factor C1s remained stable regardless of IL-1β stimulation. The complement cascade has not been studied in depth for OA, so we wanted to explore this further.

To confirm our LC-MS/MS results, we performed western blots of the supernatants, which demonstrated increased levels of intact C3 and C3's effector components (C3a, C3b, and their cleaved components) for the IL-1β treated chondrocytes (Fig. 1a). We also observed an increase in factor B as well as cleaved active factor B (Bb) in chondrocytes treated with IL-1β with no growth serum compared to control cells without growth serum (Fig. 1a). Increased levels of active factor B fragments and C3 fragments in our IL-1β treated chondrocytes compared to our control chondrocytes suggests increased complement cascade activity. The LC-MS/MS result (supplementary data 2) suggests that C1s peptide levels were relatively not affected by IL-1β stimulation in chondrocytes, so we used this as a baseline control for our western blots (Fig. 1a).

### STAT1 is required for increased C3 expression in IL-1β stimulated human primary articular chondrocytes

Since C3 showed the greatest degree of increase in IL-1β-treated cells and is the central component to the complement cascade, we decided to delve deeper into what causes the early increase in complement expression levels for IL-1β treated chondrocytes. We cross-linked biotinylated DNA from C3 gene's promoter region with nuclear proteins that bind early in the cascade and increase C3 expression. After pulling down those nuclear proteins, we disassociated them from the DNA and analyzed the proteins with LC-MS/MS (Fig. 1b). From our LC-MS/MS data (Table 1), signal transducer and activator of transcription 1 (STAT1) was increased specifically in IL-1β treated chondrocytes. As expected, a few histones were also pulled down as they are DNA-associated proteins. This result suggests that STAT1 could be responsible for the early increase in transcription of C3 from chondrocytes that were treated with IL-1β. To date, no studies have examined the relationship of STAT1, C3, and OA. Studies have found interferon signaling via the JAK1/2–STAT1 pathway was mainly the reason for increased transcription of complement genes including C3 in respiratory epithelial cells after COVID-19 infection[28]. STAT1-deficient mice have decreased C3 and factor B expression levels compared to wild type mice[29]. Therefore we hypothesize that inhibiting STAT1 could attenuate the inflammatory effects caused by the complement system that contribute to OA development.

To verify our findings, we used fludarabine, which has shown to be a specific STAT1 inhibitor in previous studies[30,31]. It is currently used clinically as chemotherapy to treat chronic lymphocytic leukemia[32]. We first assessed the toxicity of fludarabine on human primary chondrocytes via a MTS assay. For the in vitro assay, 50 μM and 100 μM fludarabine did not show substantial cellular toxicity for our primary articular chondrocytes.

After 30 min of IL-1β stimulation, C3 mRNA level started to rise with more C3 mRNA present over time (Fig. 1c). Addition of fludarabine significantly decreased C3 transcription when chondrocytes were stimulated with IL-1β at 1-h, 3-h, and 6-h. (Fig. 1c). In addition, reverse transcription quantitative real-time PCR (RT-qPCR) further confirmed the result (Fig. 1d). In the presence of fludarabine, the IL-1β treated chondrocytes produced significantly less C3 mRNA compared to the control chondrocytes with only IL-1β stimulation. These results indicate that STAT1 increases C3 transcriptional expression when stimulating human articular chondrocytes with IL-1β. Furthermore, the protein levels of C3 and factor B were also reduced by fludarabine in IL-1β treated chondrocytes (Fig. 1e). There was a substantial decrease in C3 expression with the higher dose of fludarabine treatment (100 μM) for the IL-1β cells compared to the lower dose (50 μM) of fludarabine treatment. Increased fludarabine concentration also resulted in a decrease in C3a and C3b, the cleaved, effector components for C3. These results suggest that chondrocytes are producing less C3 with fludarabine treatment. Chondrocyte supernatants were also examined as it

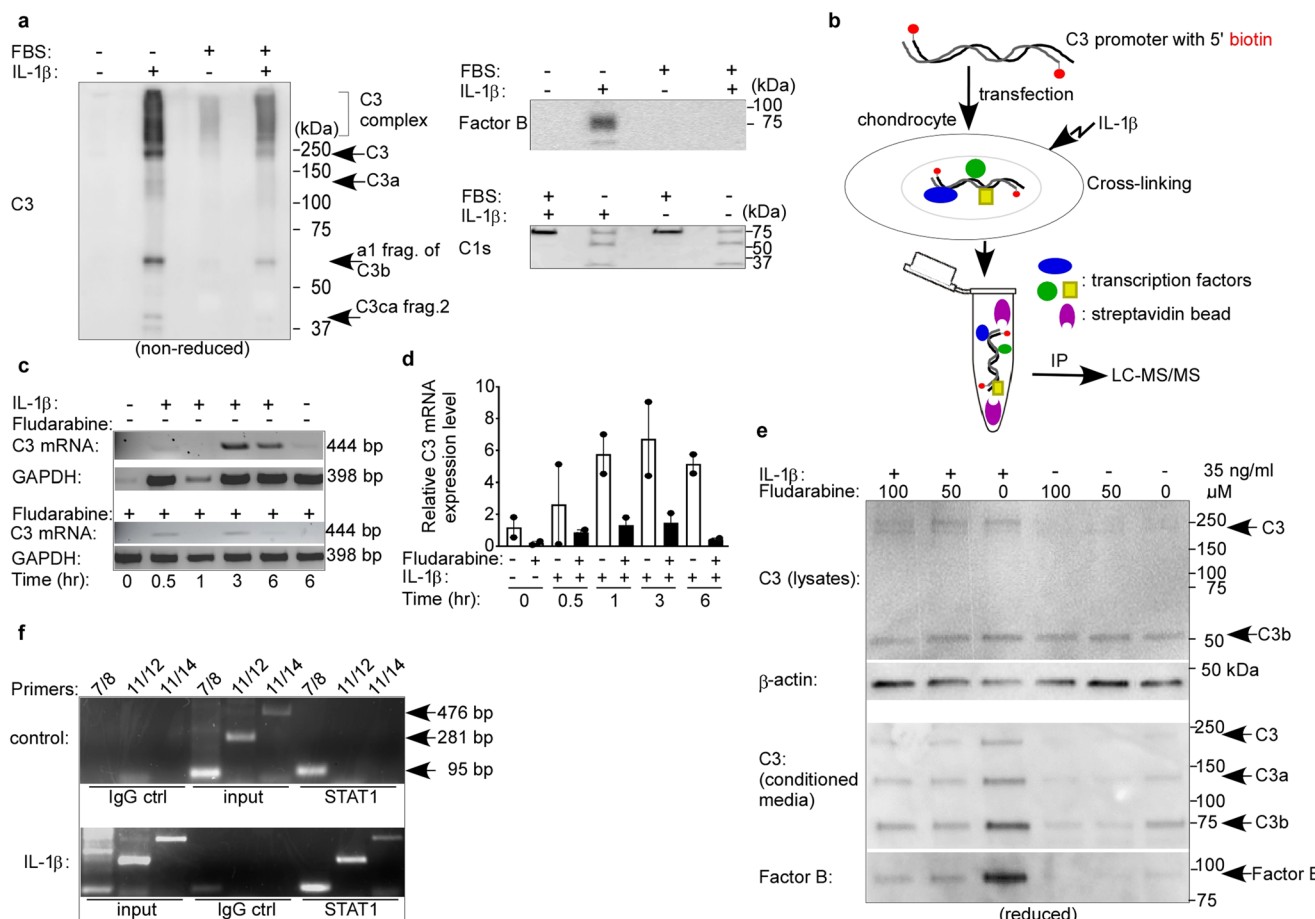

**Fig. 1 | Identification of complement 3 (C3) and its transcription activator.**
**a**, **e** The conditioned media was collected after primary human chondrocytes were stimulated with IL-1β for 66 h. Complement factors were detected with western blotting under non-reducing condition (**a**). **b** A schematic approach to identify transcription factors that increase the transcription of C3. Biotinylated C3 promoter DNA was transfected into human primary articular chondrocytes. After, these cells were stimulated with IL-1β for 30 min. Cross-linking reagent, formaldehyde, was added to cross-link DNA and proteins. The cell was lysed and the biotinylated C3

promoter-protein complexes were pulled down with streptavidin beads. The complexes were washed and cross-linking was reversed. The protein components were analyzed with LC-MS/MS. C3 mRNA transcription was affected by fludarabine (50 μM) at different time points after IL-1β stimulation by use of RT-PCR (**c**), and RT-qPCR (**d**) The error bars represent standard error of mean. **e** The protein levels of C3 and complement factor B were affected by fludarabine after IL-1β stimulation via western blotting under reduced conditions. **f** CHIP-PCR assay of transcription factor STAT1's binding region in C3 promoter. [IL-1β]: 35 ng/ml.

represents C3 levels in the intraarticular space. Increased concentrations of fludarabine inhibited C3 excretion from chondrocytes treated with IL-1β. Furthermore, there were decreased levels of active fragments of C3 with increasing fludarabine concentration. These results depict that IL-1β stimulated human articular chondrocytes produce less C3 and its active cleaved fragments with fludarabine. Together, our data suggests STAT1 can promote C3 and factor B expression in IL-1β treated chondrocytes, and fludarabine could be a good drug candidate to inhibit complement factor expression and secretion in OA.

## STAT1 association sites in the C3 promoter region
To define which regions STAT1 interacts with C3 promoter DNA, we constructed several primer pairs (listed in supplementary data 1) that cover different areas of the C3 promoter region and performed CHIP-PCR (Fig. 1f). There was increased pull down activity specifically at primer regions 11/12 and 11/14 in IL-1β treated chondrocytes by the STAT1 antibody. Furthermore, the input control depicts that the chromatin was sheared properly and had an adequate amount of DNA loaded with the primers. Collectively, this result suggests that STAT1 is associating with the C3 promoter in the region between primers 11-14, corresponding to a 476 bp region, from the 365th to the 841st nucleotide according to the C3 promoter sequence [GenBank: X62904].

To further specify and confirm the region where STAT1 is responsible for increasing C3 expression, we performed a C3 promoter luciferase assay (Fig. S1b). Previous studies have pooled together common regions of base pairs that STAT1 associates with, which were identified as consensus sequences[33]. There are two STAT1 consensus sequences based on previous studies that match at 530−539 bp and 644−650 bp of the 1081 bp C3 promoter region[33]. We mutated those regions and performed a dual-luciferase report system to determine effects on C3 expression (see Supplementary Data 1 for primers used). The result demonstrated that when both sites are mutated, there was less C3 activity compared to the intact C3 promoter. In addition, mutating only one of the consensus sequences (either 530 or 644) was not as effective as mutating both sites (Fig. S1b). As expected, the dual-luciferase results for STAT1 binding regions are consistent with the CHIP-PCR DNA regions.

## Intra-articular injection of fludarabine, a STAT1 inhibitor, improved cartilage thickness and structure of knee joints in Dunkin-Hartley guinea pigs
To assess the translational impact of our results, we performed intra-articular injections of fludarabine in Dunkin-Hartley guinea pigs, which spontaneously develop OA. Therapeutic doses were determined through a dose-finding pilot study, which demonstrated reactive synovitis starting at

**Table 1 | The nuclear proteins bound to C3 promoter after IL-1β stimulation[a]**

| Name | UniProt | IL-1β | control |
|---|---|---|---|
| Histone H4 | P62805 | 3 | 4 |
| Histone H2B type 1-B | P33778 | 3 | 1 |
| Interferon-induced GTP-binding protein | | | |
| Mx2 | P20592 | 3 | 0 |
| Interferon-induced GTP-binding | | | |
| protein Mx1 | P20591 | 3 | 0 |
| Signal transducer and activator of | | | |
| transcription 1-alpha/beta (STAT1) | P42224 | 2 | 0 |
| Histone H1.1 | Q02539 | 1 | 0 |
| Histone H2A type 1-H | Q96KK5 | 1 | 3 |
| Histone H4 | P62805 | 3 | 4 |
| Eukaryotic initiation factor 4A-II | Q14240 | 1 | 0 |
| Histone H2B type 1-B | P33778 | 3 | 2 |

[a]Unique peptides present.

250 μM in Dunkin-Hartley guinea pigs' knees. Therefore, each male Dunkin-Hartley guinea pig was assigned a dose, either 100 or 200 μM, of fludarabine. Their left legs served as a vehicle PBS injection control and their right legs served as the treatment legs.

Histological analysis showed that cartilage thickness improved with fludarabine treatment in tibial plateaus, with the medial tibia plateau showing significant changes in cartilage thickness for both fludarabine doses compared to the PBS control injections (Fig. 2a). In addition, the lateral tibial plateau showed significance in preserving cartilage thickness with 200 μM fludarabine dose. This data indicates that fludarabine can prevent cartilage loss and preserve cartilage structure in these aged guinea pigs. We performed hematoxylin and eosin (H&E) stains of the lateral and medial knee for our control legs, 100 μM, and 200 μM fludarabine treated legs (Fig. 2b) to visualize cartilage structure. Noticeably, the cartilage thickness and structure were better preserved for the fludarabine treated legs, which is consistent with objective measurements. These results suggest that fludarabine could prevent cartilage damage in spontaneous OA.

### Intra-articular injection of fludarabine improved OA features in guinea pig knees

The lateral tibial plateau showed a decrease in (bone mineral density) BMD with fludarabine treatment compared to control, with a significant difference for the 200 μM dose (Fig. 2c). For the medial tibial plateau, there was no significant difference with BMD, but there was a trend for decreasing BMD with the fludarabine treated legs. For the tibial subchondral bone, there is a significant decrease in BMD for 200 μM dose compared to control. In contrast, there were no significant differences in the tibial trabecular bone. We were able to visualize on μCT that the tibial and femoral bone structures improved with fludarabine treatments compared to control knees. There were more bony erosions as well as increased bone thickness in the control legs (Fig. 2d). In the trabecular bone, there did not seem to be a particular trend for bone volume (BV), tissue volume (TV) and bone volume/tissue volume percentage (BV/TV%), for fludarabine treated versus control legs (Fig. 2e). However, we saw a significant decrease in BV/TV% for the 200 μM fludarabine dose compared to control for the subchondral bone (Fig. 2e). As well as we observed trabecular bone pattern factor (Tb.pf) was significantly increased in the 200 μM fludarabine dose (Fig. 2f).

Proteoglycan loss was determined via toluidine blue stains for the knee joints treated with PBS control or fludarabine. The femoral condyles showed better, lower scores for the fludarabine treated joints compared to the control legs (Fig. 3a). There was a similar pattern for the tibial plateaus, which depicted significant preservation in proteoglycan content for both 100 and 200 μM fludarabine doses. Toluidine blue stains portrayed the

improved proteoglycan content, as well as cartilage structure and thickness for the fludarabine treated legs at 100 and 200 μM fludarabine doses compared to controls (Fig. 3b), which also matched our analyses and the H&E stains. The modified Mankin scores were consistent with the proteoglycan scores (Fig. 3c). There is improved Mankin score for the femur condyles and tibial plateaus with significantly lower scores for the medial tibial plateau for 100 and 200 μM fludarabine. These results suggest that fludarabine helps protect against proteoglycan loss and cartilage structure damage in spontaneous OA.

### Discussion

Over the past decades, the understanding of OA pathogenesis has expanded from OA being a noninflammatory, 'wear and tear' disease to whole joint pathology featuring synovitis, cartilage damage, subchondral bone remodeling, and osteophyte formation[34]. OA pathology involves a variety of factors, such as mechanical loading, aging, inflammation and metabolic changes, and the activation of different signaling pathways[35]. Maintaining chondrocyte balance and reducing cartilage degeneration are critical to alleviating OA development and progression[36].

To date, all currently approved drugs are for symptom control and no effective treatment exists to prevent the progression of osteoarthritis. Despite the progress in studying OA pathogenesis, the etiology and pathological mechanisms of OA are not yet fully understood[34,37]. In this study, we identified C3 and STAT1 as novel molecular targets in the pathogenesis of OA, such as cartilage degradation and bony structural changes. This opens the path for more targeted and effective therapeutic interventions as seen in our use of fludarabine to effectively attenuate C3 expression and OA progression.

It is established in OA pathogenesis that chronic, low-grade, localized inflammation of the knee joint aggravates patient symptoms, causing more cartilage damage, and accelerates disease progression[34]. Several inflammatory mediators such as IL-1β and TNF-α, and reactive metabolites such as reactive oxygen and nitrogen species reportedly contribute to the onset and progression of OA[27]. Previous studies have examined synovial fluid of patients with osteoarthritis and found elevated IL-1β and complement proteins which match our in vitro studies[4,7,24]. Since the complement cascade can amplify itself, uncontrolled activation can cause devastating effects, such as attacking the host cells. Deposition of C3 and other complement factors have been reported in the osteoarthritic cartilage of both animal models and OA patients[7,38]. Previous studies have also determined that increased local synovial production of C3 positively correlates with the severity of pathological grading of primary OA disease of the knee joint, rather than systemic complement C3[7]. The exact mechanism of complement deposition in synovium and cartilage in the setting of OA and the role of the complement cascade in OA pathogenesis still remains to be determined[34]. These studies also show that both synovial C3a and C3b levels significantly correlated with histologic severity of arthritis, but do not correlate with systemic blood inflammatory markers (ESR or CRP levels) or blood plasma C3a and C3b levels, suggesting no link between systemic and local factors of complement factors produced by the joint, which is expected as the joint space is an isolated environment[7]. Here, we also demonstrated that human chondrocytes are able to produce local complement factors in their active forms, which contributes to the inflammation in OA.

Activation of the complement pathway (via classical, alternative, and mannose-lectin) results in the formation of C3 convertase, which cleaves C3 into its effector components. C3a is involved in inflammatory effects while C3b can activate C5 convertase, which cleaves C5 into its effector components. One of which promotes the formation of C5b-9 as a MAC[39]. This complex forms pores on pathogen/target cells, leading to osmolysis and cell death. Studies have shown that C5⁻/⁻ null mice develop significantly less trauma-induced OA (through meniscectomy or destabilization of meniscus), and do not develop collagen-induced arthritis (CIA), a model for rheumatoid arthritis (RA)[40,41]. Aging (or spontaneous) OA is different from trauma-related OA or RA. In human primary chondrocytes, we did not detect complement C5 in the culture media. We also

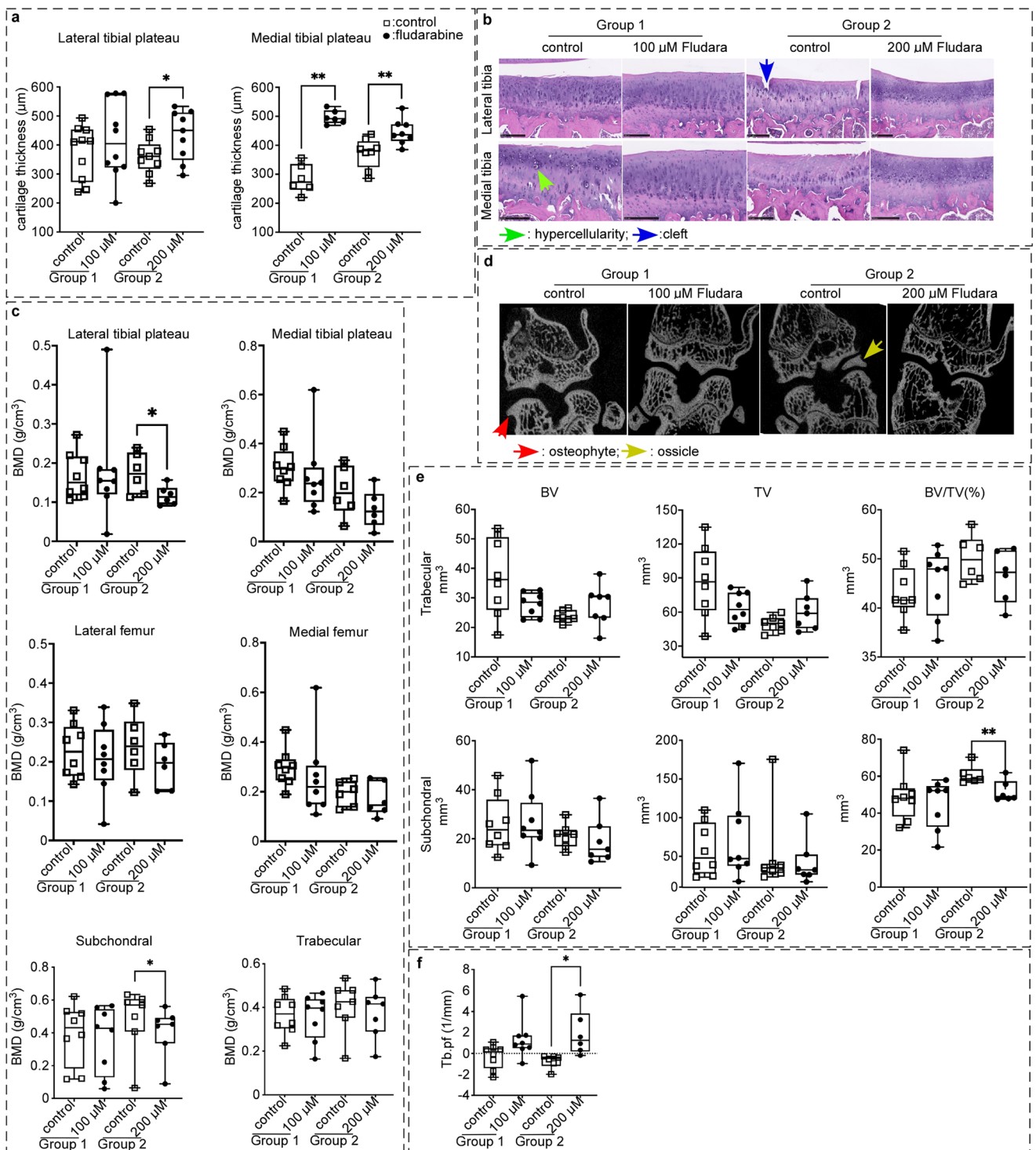

**Fig. 2 | The effect of STAT1 inhibitor, fludarabine, on the pathogeneis of aged, osteoarthritic Dunkin-Hartley guinea pigs.** 5-month-old male Dunkin-Hartley guinea pigs were given one of two different doses of fludarabine injections (100 μM and 200 μM) every 7−10 days for 7-months. **a** Cartilage thickness for lateral and medial tibial plateaus was determined at 5x microscopy and measured using NDP view 2 software. Significance was determined by paired two-tailed *t* test. Lateral tibia 200 μM ($n = 9$, $p = 0.0227$, $t = 2.815$, df = 8). Medial tibia 100 μM ($n = 6$, $p = 0.0012$, $t = 6.565$, df = 5), 200 μM ($n = 8$, $p = 0.0041$, $t = 4.188$, df = 7). **b** Hematoxylin and eosin stains of cartilage features of fludarabine treatments compared to vehicle controls. **c** Bone mineral density (BMD) was measured based on μCT images.

Significance was determined by a paired two-tailed sample *t* test. Lateral tibia plateau 200 μM ($n = 6$, $p = 0.0154$, $t = 3.606$, df = 5). Subchondral 200 μM ($n = 7$, $p = 0.0286$, $t = 2.865$, df = 6). **d** Radiographic bony features of fludarabine treatments compared to vehicle control using μCT. **e** Bone volume (BV)/tissue volume (TV) in sub-chondral and trabecular bone was analyzed with a paired two-tailed sample *t* test. Subchondral BV/TV 200 μM ($n = 6$, $p = 0.0030$, $t = 5.367$, df = 5). **f** Trabecular bone pattern factor (Tb.pf) of the tibia was analyzed through paired two-tailed sample *t* test. 200 μM fludarabine ($n = 6$, $p = 0.0257$, $t = 3.141$, df = 5). The black scale bar in (**b**): 250 μm. All points are plotted on a box plot. Control: injection of equal volume of PBS; *$p < 0.05$, **$p < 0.01$.

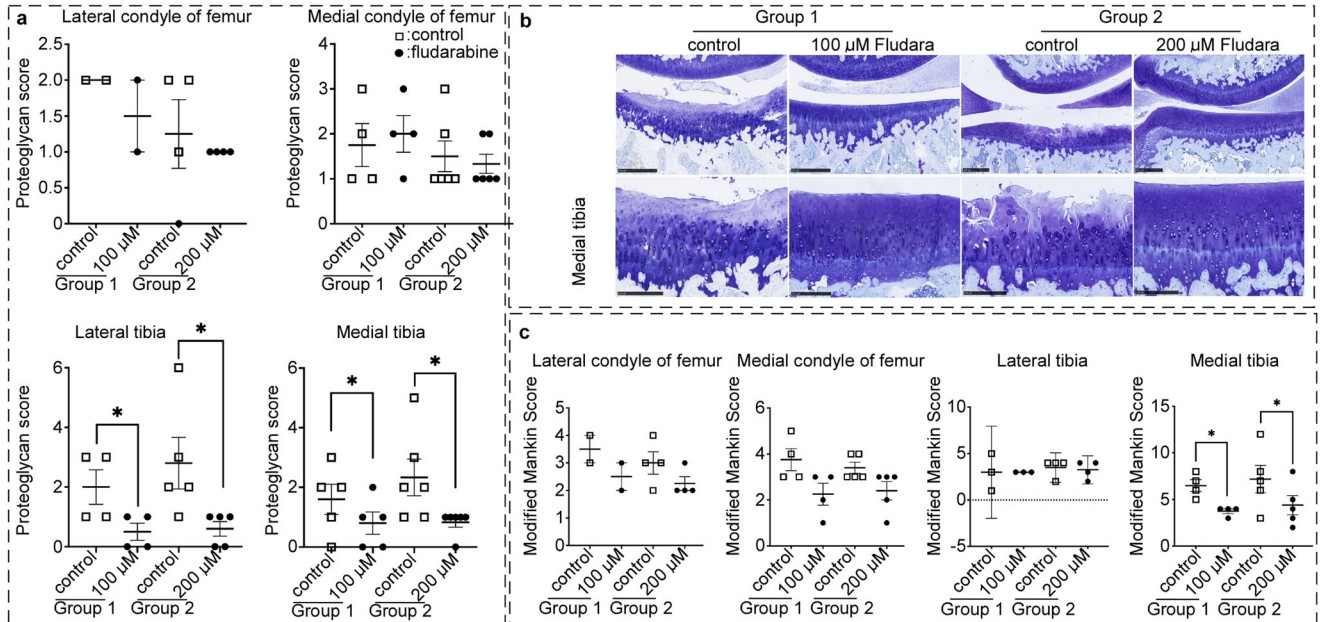

**Fig. 3 | The osteoarthritis lesions of knee joints were histologically analyzed in aged, osteoarthritic Dunkin-Hartley guinea pigs with vehicle control versus fludarabine intra-articular injection. a, c** The score of proteoglycan loss and modified Mankin score regarding osteoarthritis severity were calculated based on osteoarthritis scoring system. Modified Mankin score was calculated by adding together individual scores of cartilage structure, cellularity, tidemark, and osteophyte formation at 20x. Lateral tibia proteoglycan score 100 μm ($n = 4$, $p = 0.0138$, $t = 5.196$, df = 3); 200 μm ($n = 5$, $p = 0.0402$, $t = 2.994$, df = 4). Medial tibia proteoglycan score 100 μm ($n = 5$, $p = 0.0161$, $t = 4.000$, df = 4); 200 μm ($n = 6$, $p = 0.0446$, $t = 2.666$, df = 5). Medial tibia modified Mankin score 100 μm ($n = 4$, $p = 0.0105$, $t = 5.745$, df = 3); 200 μm ($n = 5$, $p = 0.0189$, $t = 3.810$, df = 4). **b** Toluidine blue stained images were used to determine proteoglycan content, as well as verify cellularity and tidemark at 20x microscopy. The black scale bar in (**b**): 500 μm for top panels, and 250 μm for lower panels. All points were plotted as individual values with error bars representing mean with SEM. Control: injection of equal volume of PBS; *$p < 0.05$.

performed C5 western blot and did not see any C5 signal. In addition, complement C3 is the most abundant in explants of cartilage and synovium tissue of OA patients while C5 or MAC activity is not detectable[42,43]. Taken together, C5 activation may not be indispensable in initiation and early development of spontaneous OA, but it would be valuable to further explore the relationship between C3 and C5 during the process of cartilage damage in spontaneous OA.

Normally, the joint is an isolated environment where substances can only enter and exit via diffusion, so complement factors in the blood will not overwhelm the joint space. Because of this, fludarabine's dosage will only need to inhibit local complement being produced in the joint capsule compared to inhibiting the larger amount of complement that is present in the blood. We did not see signs of systemic side effects in the guinea pigs with intra-articular injection of fludarabine, such as weight loss, neutropenia, or anemia. In addition, we performed TUNEL (terminal deoxynucleotidyl transferase dUTP nick end labeling) staining in the guinea pig knee joint cartilage tissue sections. Our result did not show that fludarabine increased apoptosis of chondrocytes in the cartilage tissue compared to controls. We did not perform specific gait analysis for our study, but we observed our guinea pigs from 5 to 12 months-old during which we did not observe any specific or obvious changes in their gaits or abnormal movement of their legs between controls or treated legs. This is likely due to the guinea pigs being in their early stages of knee osteoarthritis. Primary (spontaneous) osteoarthritis (OA) develops in Dunkin-Hartley guinea pigs at 4 months of age[44]. At 9 months of age, these guinea pigs have less mobility problems, but the later stages (over 18 months of age), OA becomes severe and causes up to 50% decreased mobility[45].

STAT1 can be activated by various inflammatory factors such as IL-6 and IFNs[46,47]. STAT1 is a transcription factor that promotes Fas, TNF-Related Apoptosis Inducer Ligand, and caspases, all of which are involved in cell death[46]. In addition, STAT1 promotes transcription of Interferon-stimulated gene factor 3 (ISGF3), which can feedback into increased IFN signaling[47,48].

By use of ingenuity pathway analysis (IPA) and publicly available ChIP-Seq of STAT1, interferon (INF)-JAK1/2-induced STAT1 signaling pathway is predicted to induce complement C3 expression and its activation in COVID-19 infected human lung epithelial cells[28]. Our results indicate for the first time that STAT1 is involved in promoting C3 expression in primary human articular chondrocytes. For the luciferase assay result, mutating two consensus sequences of STAT1 showed the most decrease in C3 production, compared to only mutating one of the sites individually. This is consistent with previous studies that have found STAT1 acts via cooperative binding to enhance activity[49]. Cooperative binding is when transcription factors bind to multiple different regions of DNA to enhance transcription. It has been determined that STAT1 cooperatively binds to different inflammatory genes as well as mathematical modeling showed that cooperativity allows wild-type STAT1 to function at more than tenfold reduced concentration compared with the cooperativity-deficient mutant[49,50].

In *Stat1*[-/-] null mice, bone mass is increased and the healing of a fractured bone (callus remodeling and membranous ossification) is also accelerated. This is due to the increased differentiation of osteoblasts in *Stat1*[-/-] mice. Two transcription factors, Runx2, and *Osterix* (Osx) are involved in the STAT1 signaling pathways[51,52]. In addition, inhibition of STAT1 by fludarabine increases ossification process[52]. In chondrocytes, a RUNX family member, RUNX1 is critical in stimulating cell proliferation and maintaining joint cartilage integrity[53]. Therefore, it would be useful to further explore the relationship of STAT1 and RUNX1 signaling in chondrocytes and OA development in future studies.

These preliminary studies led us to hypothesize that mitigating local C3 production in the joint space with a STAT1 inhibitor is a desirable therapeutic strategy. Fludarabine has already been approved for use as a chemotherapy agent and our studies have shown a potential use of repurposing this drug to treat OA. Using fludarabine in vitro decreased complement factor 3 expression as well as factor B at the transcription and translation level. Furthermore, giving fludarabine in vivo showed significant improvement in bony and cartilage microstructure on μCT and histology in

aged guinea pigs suggesting the efficacy of fludarabine for treating spontaneous OA. BMD tends to increase in Dunkin-Hartley guinea pigs over time as OA develops starting at 1 month[54,55]. In addition, there is increased BV/TV% in older Guinea pigs[54,56]. Fludarabine treatment showed a significant decrease in BMD compared to the control injections for the 200 μM dose, as well as a significant decrease in BV/TV%. For aged, osteoarthritic guinea pigs there is decreased Tb.pf[54]. Our results showed increased, improved Tb.pf for fludarabine treated knees. Thus, these results suggest that the specific STAT1 inhibitor, fludarabine, can significantly improve upon the bony and cartilage microstructure of aged Dunkin-Hartley guinea pigs and help preserve bone and cartilage integrity by inhibition of the complement pathway.

Taken together, our findings indicate STAT1 is involved in primary osteoarthritis pathogenesis, and fludarabine could potentially prevent the progression of OA by targeting complement inflammatory pathways. To our knowledge, this is the first study to demonstrate that inhibition of STAT1 using fludarabine can attenuate C3 hyperexpression and protect knee joints in aged guinea pigs. Our results provide rationale for clinical trials using fludarabine to treat OA in aging patients as a promising disease-modifying drug candidate for treatment of spontaneous OA.

## Methods

### Cell culture and in vitro osteoarthritis (OA) stimulation

Human chondrocytes were purchased from Cell Applications, San Diego CA, USA (Catalog #: 402K-05a). Complete media, trypsin, chondrocyte growth supplement, and basal media without any growth factors were also purchased from Cell Applications (Catalog #: 402K-05a). Chondrocytes were verified by performing Alcian blue stain and never surpassed 6 passages to make sure they do not dedifferentiate into fibroblast.

Human chondrocytes were cultured in a T-75 flask (Greiner, Catalog #: 658175) until 90−95% confluency. Chondrocytes were seeded in a 6 well plate at a confluency of $5 \times 10^5$. All cells were cultured in a humidified atmosphere containing 5% $CO_2$ at 37 °C. When cells reached 80% confluency, it was switched to basal media without any growth factors for 4 h. 10 or 35 ng/mL of IL-1β (Stem Cell Technology, Vancouver, CA) was added for 66 h per manufacturer instructions. Supernatant was collected for ELISA and LC-MS/MS analysis. The cell lysates were used for western blots.

### ELISA (enzyme-linked immunosorbent assay)

Human articular chondrocytes were grown in 6-well plates. 100 μL of the supernatant was assessed using commercially available ELISA kits, according to the manufacturer's instructions (IL-6 ELISA MAX Deluxe kit Biolegend, San Diego CA, USA). Total IL-6 was measured with the following reagents: anti-human IL-6 (Biolegend; 430506; 1/200), human IL-6 standard (Biolegend; 430506), and rat monoclonal anti-human IL-6 detection antibody (Biolegend; 430506; 1/200). Linear regression analysis or measurement of absorbance was used to determine concentration of IL-6.

### Western blots

Tissue and cells were extracted in RIPA buffer (1% NP-40, 0.5%NaDOC, 0.1% SDS, 2 mM EDTA (pH8.0), 150 mM NaCl, 50 mM Tris-HCl (pH 8.0)), which was supplemented with protease inhibitor cocktail (Thermo Fisher). 4x Laemmli buffer (Biorad) was added. Beta-mercaptoethanol was only added for reducing conditions. For the fludarabine treated cells, cells were pretreated with fludarabine for 12 h. Then, IL-1β was added for 24 h. Protein concentrations were determined using a BCA protein assay kit (Bio-Rad, 5000112). Samples were incubated at 95 °C for 5 min. The protein lysates were resolved by Bio-Rad 4–15% precast gradient polyacrylamide gel. Proteins were transferred to nitrocellulose membrane. After protein transfer, the membrane was blocked with 5% skim milk for 1 h at room temperature and incubated with antibodies overnight at 4 °C. After three washes with TBST, the membranes were incubated with HRP-conjugated secondary antibody at room temperature, and then washed three times with TBST. Membrane was incubated with Thermo Scientific SuperSignal West Dura Extended Duration Substrate for 3 min. The images were captured

with a chemiluminescence detection system (Azure Biosystems). After imaging, the membranes were stripped with Restore Plus for 10 min shaking at RT (Thermo Fisher). The primary antibodies employed were as follows: human anti-C1s (1:2000, AF2060SP, Fisher Scientific), human anti-Factor B (1:2000, AF2739, R&D systems), human anti-complement 3 (1:1000, PA129715, Thermo Fisher), human anti-β-actin (1:1000, 3700, Cell signaling). The secondary antibodies employed were as follows: Rabbit anti-sheep, HRP (1:10,000 PI31480, Thermo Fisher), Goat anti-Mouse IgG (H + L) Secondary Antibody, HRP (1:800, 32430, Thermo Fisher), Rabbit anti-Goat IgG (H + L) Secondary Antibody, HRP (1:10,000, 31402, Thermo Fisher).

### RT-PCR

Chondrocytes were seeded in a 6-well plate at a confluency of $1.0 \times 10^6$ and grown to 95% confluency. 50 μM fludarabine (Selleck Chemicals) was added to the respective cells grown in basal media. After 12 h with just fludarabine treatment, 35 ng/mL of IL-1β was added for the respective time points. After each time point, RNA was isolated using RNAeasy Mini kit (Qiagen) per manufacturer's instructions. The OneTaq RT-PCR Kit (New England Biolabs) per manufacturer's instructions was used to isolate the cDNA from our isolated RNA. PCR with Phusion Plus DNA Polymerase (Thermo Scientific) was performed using primers listed in supplementary data 1. RT-qPCR reactions were performed in duplicates using SYBR green qPCR supermix (Bio-Rad). Transcripts were quantified using the following program: 95 °C 30 s, followed by 40 cycles of 95 °C for 15 s, 60 °C for 1 min (Applied biosystems, Thermo Fisher). Relative level of each transcript was normalized to expression levels of GAPDH (glyceraldehyde 3-phosphate dehydrogenase) by use of the 2-ΔΔCt method[57]. The primers are listed in the supplementary data 1.

### SDS-PAGE and Coomassie Blue Staining for LC-MS/MS

Supernatant for cells grown described above for in vitro OA stimulation were concentrated using Amicon Ultra-0.5 Centrifugal Filter Unit 10 kDa (Millipore Sigma). Then 4x Laemmli Sample buffer (Bio-Rad) with Beta-Mercaptoethanol was added to the concentrated proteins in the supernatant and boiled at 95 °C for 5 min. After, samples were run on Bio-Rad 4–15% precast polyacrylamide gel until the protein ladder was separated out to 0.7 cm. The gel fragments were excised and put in Gel-fixing solution (50% ethanol in dd$H_2O$ with 10% acetic acid) for 25 min shaking at room temperature. The gel was washed with dd$H_2O$ twice while shaking at room temperature for 5 min each. Then the gel was stained with Coomassie Blue (0.5% Coomassie Blue G-250, 50% methanol, 10% acetic acid) for 8 min at room temperature with shaking. Then the gel was washed with dd$H_2O$ for 2 h at room temperature with shaking, changing the dd$H_2O$ 4 times. Then the gel was destained (50% methanol in dd$H_2O$ with 10% acetic acid) shaking at room temperature shaking for one hour, while changing the solution 3 times. The gel continued to destain the gel overnight at 4 °C with shaking. The next day the gel bands were sliced with a clean razor and then stored in 1.5 mL microcentrifuge tubes with storage solution (5% acetic acid in dd$H_2O$) and kept at 4 °C until they are ready to send out for LC-MS/MS analysis. The mass spectrometry analysis was performed at Taplin Biological Mass Spectrometry Facility (Harvard Medical School).

Co-Immunoprecipitation (Co-IP) Assay for C3 promoter and transcription factor complexes Complement factor 3 promoter region was amplified using Platinum Taq DNA Polymerase High Fidelity (Thermo Fisher), and the primers are listed in supplementary data 1. Each primer has a biotin tag on the 5' end. PCR product was resolved on a 0.70% Agarose gel and purified the gel fragment with Gel Extraction kit (Qiagen) per manufacturer's instructions. Then the fragments were confirmed with DNA sequencing. Chondrocytes were transfected with these fragments for 20 h using Lipofectamine 3000 (Thermo Fisher) per manufacturer's instructions. After 20 h of transfection, cells were grown with complete media for 3 h to recover from transfection. After 3 h, cells were serum starved for 1 h. Then, 35 ng/mL IL-1β was added for 45 min. Then cells were cross-linked with 1% formaldehyde made fresh for 15 min at room temperature. We aspirated the

media then added 125 mM glycine in molecular grade water for 5 min at room temperature to quench the cross-linking. Then 500 μL of CHIP lysis buffer (1% TritonX-100, 0.1% NaDOC, 0.1% SDS, 1 mM EDTA (pH8.0), 140 mM NaCl, 50 mM Tris-HCl (pH 8.0)) with fresh proteinase inhibitor (Thermo Fisher) was added to the cells and the cells were scraped into a 1.5 mL microcentrifuge tube. The mixture was shaken on ice for 15 min. Then 500 μL of RIPA lysis buffer (1% NP-40, 0.5%NaDOC, 0.1% SDS, 2 mM EDTA (pH8.0), 150 mM NaCl, 50 mM Tris-HCl (pH 8.0)) with fresh proteinase inhibitor (Thermo Fisher) was added and shaken on ice for 1 h. The mixture was centrifuged at $14,500 \times g$ for 10 min at 4 °C and the supernatant was transferred into new 1.5 mL microcentrifuge tube. The control agarose beads were pre-cleared and all the supernatant was added the into Control Agarose Resin (26150, Thermo Fisher) and shook at 4 °C for 1 h. The beads were then centrifuged $2000 \times g$ at 4 °C for 1 h and the supernatant was transferred to NeutrAvidin agarose (Thermo Fisher) and shaken at 4 °C overnight. After, beads were washed with 1 mL RIPA buffer twice, 1 mL RIPA buffer +0.3 M NaCl twice, 1 mL LiCl buffer twice (0.25 M LiCl, 1% NP-40, 1% NaDOC, 1 mM EDTA, 10 mM Tris-HCl (pH 8.0)), with shaking for 10 min at 4 °C and centrifuge 3000 RPM for 1 min each step. After, 25 μL of the solution was transferred into a new 1.5 mL microcentrifuge tube for PCR analysis to confirm transfection. The rest of the supernatant was centrifuged at $23,000 \times g$ for 1 min at room temperature.

25 μL from the Co-IP above was washed with TE buffer (1 mM EDTA, 10 mM Tris-HCl (pH 8.0)) for 10 min shaking at 4 °C and centrifuge $1000 \times g$ for 1 min. Agarose beads were resuspended with TE buffer and immediately used for PCR analysis with PCR SuperMix (12532016, Thermo Fisher) with primers listed in supplementary data 1.

The rest of the supernatant was transferred to a new tube with 500 μL of 1 x Laemmli sample buffer (62.5 mM Tris-HCl, 0.07 M SDS, pH 6.8.) 95 °C for 20 min to reverse formaldehyde cross-links. They were centrifuged at 2500 g for 1 min and the supernatant was moved into an Amicon Ultra-0.5 Centrifugal Filter Unit 10 kDa (Millipore Sigma). Then the samples were run on an SDS-PAGE gel and stained with Coomassie blue for LC-MS/MS analysis the same way as mentioned above.

## Chromatin immunoprecipitation (ChIP)
Cells were seeded at $3 \times 10^6$ cells in 100-mm petri dish. When cells reached confluency, cells were starved with no serum for 1 h. Then 35 ng/mL of IL-1β was added for 45 min. After, 1% formaldehyde solution was added at room temperature for 9 min 30 s into each dish for cross-linking. Then cells were quenched with 125 mM Glycine for 5 min. 0.25% Trypsin/EDTA was added into each 100-mm dish until all cells are detached and the cells were counted. Then ice-cold PBS with 0.5 mM PMSF was added into each dish. Cells were centrifuged at $2000 \times g$ for 5 min at 4 °C. Then the cell pellet was suspended in hypotonic buffer (10 mM Tris-HCl, pH 7.4, 3 mM MgCl2, 10 mM NaCl, and 0.1% IGEPAL CA-330) with fresh proteinase inhibitor (Thermo Fisher) on ice for 20 min, with agitation every 5 min. The tube is centrifuged at $2000 \times g$ for 5 min at 4 °C. and the supernatant was transferred into a new tube. The nuclei pellet is suspended and dissolved in MNase digestion buffer with fresh proteinase inhibitor (Thermo Fisher), then centrifuged at $2000 \times g$ for 5 min at 4 °C. Cell nuclei pellet was resuspended in 500 μL of MNase digestion buffer fresh proteinase inhibitor. 0.42 μL per $4 \times 10^6$ cells of Micrococcal Nuclease (10011, Cell Signaling) was added. The tube was inverted several times and incubated for 20 min at 37 °C with frequent mixing every 3–5 min to digest DNA to lengths of ∼150−900 bp. Digestion was stopped with 0.5 M EDTA and placing on ice for 2 min. The digested DNA was run on a gel to confirm proper shearing of chromatin. Pellet nuclei were collected by centrifugation at maximum speed, $21,000 \times g$ for 1 min at 4 °C. ChIP lysis buffer (1% TritonX-100, 0.1% NaDOC, 0.1% SDS, 1 mM EDTA (pH8.0), 140 mM NaCl, 50 mM Tris-HCl (pH 8.0)). with fresh protease inhibitor was added into the pellet nuclei and shook on ice for 1 h. Then it was observed under the microscope to make sure the nuclei were broken. Then it was centrifuged at $9400 \times g$ in a microcentrifuge for 10 min at 4 °C. Digested chromatin was diluted into

ChIP dilution buffer (16.7 mM Tris-HCl, pH 8., 167 mM NaCl, 1.2 mM EDTA, 0.01% SDS, 1.1% Triton X-100) with fresh proteinase inhibitor (Thermo Fisher) in a 1:4 dilution. The chromatin solution was added to Control Agarose Resin (26150, Thermo Fisher) and shook at 4 °C for 2 h to preclear the chromatin solution. Then we split the solution into aliquots, as well as a fraction for input control. All aliquots were added to Protein A/G Plus Agarose (26159, Thermo Fisher). Then, one aliquot was added to STAT1 Rabbit Antibody (Catalog #: 9172S, Cell Signaling). Another aliquot was added to Normal Rabbit IgG antibody (Catalog #: 2729S, Cell Signaling) as a negative control. Each aliquot was shaken overnight at 4 °C. The beads were washed two times in low salt buffer (20 mM Tris-HCl, pH 8.1, 150 mM NaCl, 2 mM EDTA. 0.1% SDS, 1% Triton X-100) with fresh proteinase inhibitor (Thermo Fisher), washed once with high salt buffer (20 mM Tris-HCl, pH 8.1, 2 mM EDTA, 500 mM NaCl, 0.1% SDS, 1% Triton X-100.), washed once with 1 mL of LiCl wash buffer (10 mM Tris, pH 8.0, 0.25 M LiCl, 1% NP-40, 1% deoxycholate, 1 mM EDTA), washed once with TE buffer (1 mM EDTA, 10 mM Tris-HCl (pH 8.0)). The samples were shaken at 4 °C for 4 min, then centrifuged at $3400 \times g$ for 1 min. ChIP Elution buffer (1% SDS from 10% stock solution, 0.1 M NaHCO3) was added to the beads and incubated at 65 °C, for 30 min with shaking. The tubes were centrifuged at $10,000 \times g$ for 1 min. RNase A (10 mg/ml), 0.3 M NaCl, and proteinase K were added to each tube, and incubated at 65 °C for 4 h with shaking. DNA was purified using PCR purification kit (Qiagen) per manufacturer's protocol. PCR primers for ChIP DNA is in supplementary data 1 using Taq Polymerase (Catalog #: 11304011, Thermo Fisher).

## Luciferase assay
To predict STAT1 binding targets on C3, we identified consensus sequences for STAT1 that have been pooled together from previous studies. Based on these consensus sequences, site-specific mutations were cloned via PCR into the pMCS-Cypridina Luciferase vector (Catalog #: PI16149, Thermo Fisher). We also had a constitutively active pCMV-Red Firefly luciferase vector (Catalog #: PI16156, Thermo Fisher) as a control. Both vectors were co-transfected into human primary articular chondrocytes (Cell Applications) in a 24-well plate in triplicate using Lipofectamine 3000 (Invitrogen). Cells were lysed 24 h after transfection, and luciferase activity was determined by a Dual-Luciferase Reporter Assay (Catalog #: PI16183, Thermo Fisher). Cypridina luciferase activity was normalized to pCMV-Red Firefly luciferase activity. The sequences of the PCR primers used to introduce site-specific mutations are listed in supplementary data 1.

## MTS assay
Chondrocytes were cultured in 48 well plate with 300 μL of complete media. Fludarabine concentrations were added as follows: 0, 5, 10, 20, 30, 50, 100, 150, 200 and 300 μM. Each concentration was plated 4 times. After 2 h, 4 h, 1 day, 2 days, 4 days, 60 μL of MTS solution containing PES to each well (final concentration of MTS is 0.33 mg/ml). Absorbances were recorded at 490 nm based on Cell Viability Assays on the SpectraMax iD5 (Molecular Devices, USA)[58].

## Guinea pig housing
Five-month-old male, intact Dunkin-Hartley guinea pigs were purchased from an AAALAC-accredited commercial vendor and housed in an AAALAC-accredited facility in accordance with the Guide for Care and Use of Laboratory Animals. This primary study protocol was approved by the Medical University of South Carolina IACUC. All animals were housed in standard plastic guinea pig drawer style enclosures (24" × 32" × 10") with slotted floors in pairs or alone. Enclosures included two shelters (either plastic hut or plastic igloo) and one enrichment manipulandum rotated weekly (nylon bone, wooden block, or small ball). Animals had free access to timothy hay (Kaytee), Certified Guinea Pig diet #5026 (LabDiet), and reverse osmosis filtered water. A rotating supply of vegetables were supplemented approximately 5 times per week. Sanitization schedules were as follows: enclosures every other week, accessories weekly, and cage pans with noncontact paper chip bedding three times per week. Macroenviroment

parameters were as follows: 12:12 light:dark cycle, temperatures between 68 and 79 °F, and 30−70% humidity.

### Intra-articular knee injections in Dunkin-Hartley Guinea pigs

Every 7−10 days we injected guinea pigs' knees intra-articularly with 50 μL of fludarabine in the right leg (SelleckChem, Houston, TX, USA) and 50 μL of phosphate-buffered saline in the left leg (Cytiva HyClone, Logan, UT, USA). We injected half of the guinea pigs with 100 μM concentration of fludarabine into the right knee and equal volume of vehicle control, PBS, in the left knee. The other half would receive 200 μM of fludarabine on the right knee and PBS in the left knee. After 7 months, we harvested the joints for analysis, when the guinea pigs reached 1-years-old.

### Micro-computed tomography (μCT)

Guinea pig knees were excised midway down the femur shaft and above the ankle and fixed in neutral-buffered formalin (10% in phosphate buffer. Fisherbrand, Pittsburgh, PA, USA) as close to a 180° angle as possible. After 3-days, the joints were placed in containers that could fit into the μ-CT machine for scanning. Samples were scanned (SkyScan 1176 micro-CT scanner, Bruker, Kontich, Belgium) at 18 microns with an Aluminum and Copper filter, with rotation step 0.7° for 360° and offset camera. The joints were reconstructed using NRecon software (Bruker, Belgium). After reconstruction, scans were oriented in the same plane using DataViewer (Bruker MicroCT, Kontich, Belgium). Images were analyzed using CTAn (Bruker MicroCT, Kontich, Belgium). 2-D images were generated using DataViewer (Bruker, Belgium).

### Histology

After μ-CT experiments, samples were placed in formic acid decalcification solution (5% in PBS) for 8 days, changing the solution every day. The knee joints were cut into 3 segments and put in cassettes. After, they were rinsed with water and neutralize with diluted ammonia solution (4% in distilled water) for 30 min. The specimen was washed with distilled water and put in neutral-buffered formalin until they were ready for paraffin embedding. Tissues were dehydrated, embedded in paraffin using the automatic Leica TP 1020: 70% ethanol for 1 h 15 min, 95% ethanol for 1 h 15 min, 95% ethanol for 1 h 15 min, 100% ethanol for 1 h 15 min, 100% ethanol for 1 h 15 min, 100% ethanol for 1 h 15 min, 50% ethanol: 50% xylene for 1 h 15 min, 100% xylene for 1 h 15 min, 100% xylene for 1 h 15 min, 100% xylene for 1 h 15 min, paraffin for 1 h and 45 min, paraffin for 1 h and 45 min. The samples were sectioned into 5 μm slides for Hematoxylin–Eosin staining (The Medical University of South Carolina Histology Core, Charleston, SC). We also performed toluidine blue staining using 0.04% Toluidine blue. Pictures of stained slides were taken using Hamamatsu Nanozoomer Whole-slide imager (Shizouka, Japan). Samples were evaluated using the Osteoarthritis Research Society International (OARSI) recommended guidelines[59]. This semiquantitative grading scheme is based on articular cartilage structure, proteoglycan content, cellularity, tidemark integrity, and osteophyte presence.

### TUNEL (terminal deoxynucleotidyl transferase dUTP nick end labeling) Staining

Unstained sectioned formalin-fixed paraffin-embedded (FFPE) tissue specimens was deparaffinized per manufacturer's recommendation (Click-iT Plus TUNEL Assay, Thermo Fisher). The tissue specimens were fixed and permeabilized by Proteinase K. Terminal Deoxynucleotidyl Transferase (TdT) reaction and Click-iT Plus reaction were performed. A Hoechst 33342 stain was also performed to stain the nuclei of cells. The slides were imaged via confocal microscopy (Zeiss).

### Statistical analysis

For all quantitative analyses, a minimum of three biological replicates were analysed. Shapiro-wilk tests were performed for normality, and the values were normally distributed, so significance was determined by Student $t$ tests for comparisons of two groups. The following values were statistically significant: $*P < 0.05$, $**P < 0.01$, and exact $P$ values are given in the figure legends. Calculations were carried out using the GraphPad Prism 10 software package. No statistical method was used to predetermine sample size; we used the maximum number of animals our facility could hold for our in vivo experiments. Experimenters were not blinded to treatments, but data analysis was carried out blindly.

### Reporting summary

Further information on research design is available in the Nature Portfolio Reporting Summary linked to this article.

## Data availability

All data supporting the findings of this study are available within the paper and its supplementary information. Source data are provided in this paper. Numerical source data for all graphs in the manuscript can be found in the supplementary information file. Additional data related to this paper are available from the corresponding author on request.

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

## Acknowledgements
We would like to dedicate this work to the late Dr. Patrick M. Woster. He helped conceive, implement, and supervise this project. Mathew J Gregoski gave us statistical guidance. Yunxin Zhang provided histology guidance. This publication was supported by the National Center for Advancing Translational Sciences of the National Institutes of Health under Grant Numbers TL1 TR001451 & UL1 TR001450 as well as the National Institute of Dental & Craniofacial Research of the National Institutes of Health under Award Number R01DE029637. The content is solely the responsibility of the authors and does not necessarily represent the official views of the National Institutes of Health. This project was also supported in part by Dr. Patrick Woster's Smartstate Endowed Chair in Medicinal Chemistry for the Center of Cancer Drug Discovery. F.Z.X. discloses support for the research of this work from Medical University of South Carolina Summer Undergraduate Research Program (SURP).

## Author contributions
J.X.X. and Y.K.P. conceived the original idea. J.X.X. and F.Z.X. performed all the in vitro experiments as well as analyzed them and implemented the figures into the manuscript. J.X.X., A.F., A.M.B., B.B. performed the in vivo studies. F.Z.X., A.F., K.L.H. helped analyze the in vivo results. J.X.X. wrote the manuscript with support from F.Z.X. and Y.K.P., all authors were given the manuscript for edits. Y.K.P. helped supervise the project.

## Competing interests
The authors declare no competing interests.
