## [Peer Review File · Communications Biology]

Reviewers' comments:

Reviewer #1 (Remarks to the Author):

This work investigate the role of C3 during the process of OA, and provide a specific STAT1 inhibitor which could be used for the treatment of OA. The manuscript was orgnized well and suitable for the readers of this journal. And I think it could be accepted after minior revision.

1. In figure 2b, it is better to provide the full plot of Hematoxylin and eosin stains of cartilage with partial enlargement which could be in accordance with figure 1d.
2. If possible, some physical data like gait analysis data could be added to prove the conclusion in other side.

Reviewer #2 (Remarks to the Author):

Review for Communication Biology; November 2023

Therapeutic Inhibition of Complement C3 in Spontaneous Osteoarthritis

This manuscript explores the role of C3 and Stat1 in a spontaneous age-related OA model in guinea pigs. The main findings of the manuscript include human chondrocytes produce C3 and degradation products after stimulation with IL-beta, C3 is transcriptionally regulated in part by STAT1 in human chondrocytes, and that use of intra-articular fludarabine, every 1-2 weeks for 7 months, can prevent spontaneous, age-related OA in a guinea pig model. Strengths of the manuscript include that it is well written and many of the conclusions are supported by the data. While complement involvement in OA is known, the novelty of the manuscript lies in the use of intraarticular injections of fludarabine to prevent age related OA. However, there are several limitations:

- 1) The transcriptional analysis of C3 mRNA and GAPDH (appears to be semiquantitative and not by real time PCR. The GAPDH bands are for the most part are saturated making it difficult to see true ratios.
- 2) For the in vitro experiments, you are treating the chondrocyte cultures with IL-1B for 66hours. When are you adding fludarabine? Is it added at the same time as the IL-1B? What is half-life of fludarabine in tissue culture?
- 3) You tested viability at only 4 hours. Viability should be tested for longer, perhaps 1, 2, 4 days. Fludarabine will also disrupt DNA so longer time points are required.
- 4) The cartilage thickness with 200uM Fludarabine is less than with 100uM and the chondrocytes are not in the nice columns anymore. Did you perform tunnel staining on the cartilage?
- 5) A paper in Genes Dev 2003 showed that loss of STAT1 leads to increased activation of RUNX2. Do you know if fludarabine is increasing RUNX2 in the chondrocytes? This could be more important than inhibition of STAT1.
- 6) Robinson in a Nature Medicine 2011 Nov 6;17(12):1674-9. showed that C5 is required for cartilage catabolism by promoting the MAC complex. C3 activation is required for C5 activation. Is the effect on cartilage really due to inability to activate C5?
- 7) Intraarticular fludarabine could also have systemic effects. Did the guinea pigs lose weight or develop anemia?

Please see our answers to each question/comment from the reviewers in the following:

Reviewer #1 (Remarks to the Author):

This work investigate the role of C3 during the process of OA, and provide a specific STAT1 inhibitor which could be used for the treatment of OA. The manuscript was orgnized well and suitable for the readers of this journal. And I think it could be accepted after minior revision.

1. In figure 2b, it is better to provide the full plot of Hematoxylin and eosin stains of cartilage with partial enlargement which could be in accordance with figure 1d (likely 2d).

We have full plots of H & E stains for all knee joints. When we sectioned the knee joints, it was difficult to get a coronal section that aligns with our micro-CT scans as the knee joint consists of soft tissue, cartilage and bone making it difficult to get perfect, clean coronal sections. Therefore, we had images where we showed overall the femur and tibia as well as the detailed structure. Fig 2d is a micro-CT scan for the entire knee bone structure before sectioning, meaning we were able to take individual sections coronally, sagittally, and transversely through the entire joint. We did not section the knee joint the same way as we depicted the coronal images of the entire joint on micro-CT, which is why our H & E tissue samples are depicted this way in our figure.

2. If possible, some physical data like gait analysis data could be added to prove the conclusion in other side.

(The following part is included in the Discussion section of our manuscript, lines 308-315).

We did not perform specific gait analysis for our study. We observed our guinea pigs from 5- 12 months-old. During this time, we did not observe any specific or obvious changes in their gaits or abnormal movement of their legs between controls or treated legs. This is likely due to the guinea pigs being in their early stages of knee osteoarthritis. Primary (spontaneous) osteoarthritis (OA) develops in Hartley guinea pigs at 4 months of age (Jimenez et al., 1997). At 9 months of age, these guinea pigs have less mobility problems. At the later stages (over 18 months of age), OA becomes severe and causes up to 50% decreased mobility (Santangelo et al., 2014)

Jimenez PA, Glasson SS, Trubetskoy OV, Haimes HB. Spontaneous osteoarthritis in Dunkin Hartley guinea pigs: histologic, radiologic, and biochemical changes. *Lab Anim Sci.* 1997, 47(6):598-601.

Santangelo KS, Kaeding AC, Baker SA, Bertone AL. Quantitative Gait Analysis Detects Significant Differences in Movement between Osteoarthritic and Nonosteoarthritic Guinea Pig Strains before and after Treatment with Flunixin Meglumine. *Arthritis.* 2014; 503519

Reviewer #2:

This manuscript explores the role of C3 and Stat1 in a spontaneous age-related OA model in guinea pigs. The main findings of the manuscript include human chondrocytes produce C3 and degradation products after stimulation with IL-beta, C3 is transcriptionally regulated in part by STAT1 in human chondrocytes, and that use of intra-articular fludarabine, every 1-2 weeks for 7 months, can prevent spontaneous, age-related OA in a guinea pig model. Strengths of the manuscript include that it is well written and many of the conclusions are supported by the data. While complement involvement in OA is known, the novelty of the manuscript lies in the use of intraarticular injections of fludarabine to prevent age related OA. However, there are several limitations:

1) The transcriptional analysis of C3 mRNA and GAPDH (appears to be semiquantitative and not by real time PCR. The GAPDH bands are for the most part are saturated making it difficult to see true ratios.

We performed and included real time RT-qPCR result in our manuscript as Fig 1d as well as lines 156-159 (also in methods section lines 639-644), also shown here:

2) For the in vitro experiments, you are treating the chondrocyte cultures with IL-1B for 66hours. When are you adding fludarabine? Is it added at the same time as the IL-1B? What is half-life of fludarabine in tissue culture?

(We added the following information to the Methods section, lines 612-613, 634)

We serum starved the chondrocytes for 2 hours (without FBS). Then we pre-treated cultured chondrocytes (without FBS) with 50 μM and 100 μM fludarabine (Selleck Chemicals) in basal culture media for 12 hours. After 12 hours of only fludarabine treatment, 35 ng/mL of IL-1β was added for their respective time points. Then, the cells lysates and the conditioned media were collected for RT-PCR, RT-qPCR and Western blots.

Usually, half-life of drugs is not tested *in vitro* as it would need systemic clearance. Therefore, we did not perform half-life of fludarabine experiment *in vitro*. Based on reported data, the half-life of fludarabine in human is approximately 20 hours after systemic injection.

(https://cdn.pfizer.com/pfizercom/products/uspi_fludarabine.pdf)

3) You tested viability at only 4 hours. Viability should be tested for longer, perhaps 1, 2, 4 days. Fludarabine will also disrupt DNA so longer time points are required.

We performed the viability of chondrocytes *in vitro* for 1, 2, 3, and 4 days in the presence of fludarabine with MTS assay. We did not observe differences in chondrocyte viability for 1-4 days of fludarabine treatment. Since the knee joint environment is different from cultured cells, we used different concentration of fludarabine (0, 25 μ M, 50 μ M, 100 μ M, 150 μ M, 200 μ M, 300 μ M and 350 μ M), and injected into knee joint space of guinea pigs. After one and two weeks, we collected the knee samples. We observed that reactive synovitis started at 250 μ M. Therefore, we only used two concentrations of fludarabine, 100 μ M and 200 μ M as injection doses in the guinea pigs.

4) The cartilage thickness with 200uM Fludarabine is less than with 100uM and the chondrocytes are not in the nice columns anymore. Did you perform tunnel staining on the cartilage?

(The following is mentioned in discussion section lines 305-308, as well as methods (lines 821-827)

We performed TUNEL (terminal deoxynucleotidyl transferase dUTP nick end labeling) staining in the guinea pig knee joint cartilage tissue sections. The result (confocal microscopy images) did not show that fludarabine increased the apoptosis of chondrocytes compared to control. Please see the following figure (not included in the manuscript).

The TUNEL staining was performed by use of a kit from Thermo Fisher (Catalogue #: C10617). The green fluorescence is degraded nuclear DNA in apoptotic chondrocytes and blue staining is nuclear DNA by Hoechst 33342.

5) A paper in *Genes Dev* 2003 showed that loss of STAT1 leads to increased activation of RUNX2. Do you know if fludarabine is increasing RUNX2 in the chondrocytes? This could be more important than inhibition of STAT1.

(The following part is included in the Discussion section of our manuscript, lines 334-341)
In *Stat1*^{-/-} null mice, bone mass is increased and the healing of a fractured bone (callus remodeling and membranous ossification) is also accelerated. This is due to the increased differentiation of osteoblasts in *Stat1*^{-/-} mice. Two transcription factors, Runx2 (Kim et al, 2003), and Osterix (Osx) (Tajima et al, 2010), are involved in the STAT1 signaling pathways. In addition, inhibition of STAT1 by fludarabine increases ossification process (Tajima et al, 2010). In chondrocytes, a RUNX family member, RUNX1 is critical to stimulate these cell proliferation and maintain the joint cartilage integrity (Che et al. 2023). It would be useful to further explore the relationship of STAT1 and RUNX1 signaling in chondrocytes and OA development in the future.

Che X, Jin X, Park NR, Kim HJ, Kyung HS, Kim HJ, Lian JB, Stein JL, Stein GS, Choi JY. Cbfb Is a Novel Modulator against Osteoarthritis by Maintaining Articular Cartilage Homeostasis through TGF- β Signaling. *Cells*. 2023 Mar 31;12(7):1064.
Kim S, Koga T, Isobe M, Kern BE, Yokochi T, Chin YE, Karsenty G, Taniguchi T, Takayanagi H. Stat1 functions as a cytoplasmic attenuator of Runx2 in the transcriptional program of osteoblast differentiation. *Genes Dev*. 2003;15;17(16):1 979-991.
Tajima K, Takaishi H, Takito J, Tohmonda T, Yoda M, Ota N, Kosaki N, Matsumoto M, Ikegami H, Nakamura T, Kimura T, Okada Y, Horiuchi K, Chiba K, Toyama Y. Inhibition of STAT1 accelerates bone fracture healing. *J Orthop Res*. 2010; 28(7):937-941

6) Robinson in a *Nature Medicine* 2011 Nov 6;17(12):1674-9. showed that C5 is required for cartilage catabolism by promoting the MAC complex. C3 activation is required for C5 activation. Is the effect on cartilage really due to inability to activate C5?

(The following part is included in the Discussion section of our manuscript lines 284-298)
Activation of the complement pathway (via classical, alternative, and mannose-lectin) results in the formation of C3 convertase, which cleaves C3 into its effector components. C3a is involved in inflammatory effects while C3b can activate C5 convertase, which cleaves C5 into its effector components. One of which promotes the formation of C5b-9 as a membrane attack complex (MAC) (Kemper et al. 2007). This complex forms pores on pathogen/target cells, leading to osmolysis and cell death. There are also C3 independent pathways that can activate C5 (Huber-Lang et al. 2006, Wang et al. 2011). Several activated complement factors can initiate inflammatory reactions, which are involved several diseases including osteoarthritis (Markiewski et al. 2007, Silawal et al. 2018). Studies have shown that *C5*^{-/-} null mice develop significantly less trauma-induced OA (through meniscectomy or destabilization of meniscus) (Wang et al. 2011), and do not develop collagen-induced arthritis (CIA), a model for rheumatoid arthritis (RA) (Wang et al 2000). Aging (or spontaneous) OA is different from trauma-related OA or RA. Low grade inflammation and slow damage of cartilage matrix via the activation of catabolic pathways is critical for the progressive development of OA (Goldring, 2000, Schäfer et al. 2022). In human primary chondrocytes, we did not detect complement C5 that was secreted into culture media. We also performed C5 western blot in our primary human chondrocytes and did not see

any C5 signal. In addition, complement C3 is the most abundant in explants of cartilage and synovium tissue of OA patients while C5 or MAC activity is not detectable (Assirelli et al. 2020, Gobezie et al. 2007). Taken all together, C5 activation may not be indispensable in initiation and early development of spontaneous OA. However, it would be valuable to further explore the relationship between C3 and C5 during the process of the cartilage damage in spontaneous OA.

Assirelli E, Pulsatelli L, Dolzani P, Mariani E, Lisignoli G, Addimanda O, Meliconi R.

Complement Expression and Activation in Osteoarthritis Joint Compartments. *Front Immunol.* 2020; 11: 535010

Gobezie R, Kho A, Krastins B, Sarracino DA, Thornhill TS, Chase M, Millett PJ, Lee DM. High abundance synovial fluid proteome: distinct profiles in health and osteoarthritis. *Arthritis Res Ther.* 2007;9(2):R36

Goldring MB. The role of the chondrocyte in osteoarthritis. *Arthritis Rheum.* 2000; 43(9):1916-26.

Huber-Lang M, Sarma JV, Zetoune FS, Rittirsch D, Neff TA, McGuire SR, Lambris JD, Warner RL, Flierl MA, Hoesel LM, Gebhard F, Younger JG, Drouin SM, Wetsel RA, Ward PA. Generation of C5a in the absence of C3: a new complement activation pathway. *Nat Med.* 2006;12(6):682-7

Kemper C, Atkinson JP. T-cell regulation: with complements from innate immunity. *Nature Reviews Immunology.* 2007; 7:9–18

Markiewski MM, Lambris JD. The role of complement in inflammatory diseases from behind the scenes into the spotlight. *Am J Pathol.* 2007;171(3):715-27.

Schäfer N, Grässel S. Involvement of complement peptides C3a and C5a in osteoarthritis pathology. *Peptides.* 2022;154:170815

Silawal S, Triebel J, Bertsch T, Schulze-Tanzil G. Osteoarthritis and the Complement Cascade. *Clin Med Insights Arthritis Musculoskelet Disord.* 2018;11: 1-12

Wang Q, Rozelle AL, Lepus CM, Scanzello CR, Song JJ, Larsen DM, Crish JF, Bebek G, Ritter SY, Lindstrom TM, Hwang I, Wong HH, Punzi L, Encarnacion A, Shamloo M, Goodman SB, Wyss-Coray T, Goldring SR, Banda NK, Thurman JM, Gobezie R, Crow MK, Holers VM, Lee DM, Robinson WH. Identification of a central role for complement in osteoarthritis. *Nat Med.* 2011; 17(12):1674-1679.

Wang Y, Kristan J, Hao L, Lenkoski CS, Shen Y, Matis LA. A role for complement in antibody-mediated inflammation: C5-deficient DBA/1 mice are resistant to collagen-induced arthritis. *J Immunol.* 2000;164(8):4340-7

7) Intraarticular fludarabine could also have systemic effects. Did the guinea pigs lose weight or develop anemia?

(The following part is mentioned in the Discussion section of our manuscript, lines 303-305)
The guinea pigs' weights were taken weekly and we did not observe any major differences. We collected blood samples from 6 guinea pigs, and the CBC with differential cell count (white blood cells, red blood cells, hemoglobin level, and platelet number) were all normal. Therefore, we do not suspect any systemic effects of fludarabine. This aligns with the prediction that fludarabine would not reach systemic circulation and cause off-target effects likely due to an isolated environment of knee joint and the low concentration of fludarabine.

REVIEWERS' COMMENTS:

Reviewer #1 (Remarks to the Author):

The author have revised the manuscript according to my suggestions, and it could be accepted.

Reviewer #2 (Remarks to the Author):

The authors have appropriately addressed the concerns raised by the reviewers. No additional recommendations are needed.